# Successful Eradication of Porcine Epidemic Diarrhea in an Enzootically Infected Farm: A Two-Year Follow-Up Study

**DOI:** 10.3390/pathogens10070830

**Published:** 2021-07-01

**Authors:** Guehwan Jang, Jonghyun Park, Changhee Lee

**Affiliations:** Animal Virology Laboratory, School of Life Sciences, BK21 FOUR KNU Creative BioResearch Group, Kyungpook National University, Daegu 41566, Korea; wayyonim12@naver.com (G.J.); man3807@naver.com (J.P.)

**Keywords:** biosecurity, endemic infection, eradication, herd immunity, monitoring and surveillance, prime-boost vaccination, PEDV

## Abstract

Porcine epidemic diarrhea virus (PEDV) has negatively affected the welfare of animals and their productivity in South Korea for three decades. A shortage of effective control measures has led to the virus becoming endemic in domestic pig populations. This study aimed to describe how our intervention measures were implemented for PEDV elimination in an enzootically infected farm. We operated a risk assessment model of PEDV recurrence to obtain information about the virus itself, herd immunity, virus circulation, and biosecurity at the farm. Next, we conducted a four-pillar-based two-track strategy to heighten sow immunity and eradicate the virus, with longitudinal monitoring of immunity and virus circulation, involving strict biosecurity, prime-boost pre-farrow L/K/K immunization, all-in-all-out and disinfection practices in farrowing houses, and disinfection and gilt management in wean-to-finish barns. In particular, we observed a high prevalence and long-term survival of PEDV in slurries, posing a critical challenge to PED eradication and highlighting the necessity for consecutive testing of barn slurry samples and for the management of infected manure to control PEDV. Genetic analysis of PEDVs in this farm indicated that genetic drift continued in the spike gene, with a substitution rate of 1.683 × 10^−4^ substitutions/site/year. Our study underlines the need for active monitoring and surveillance of PEDV in herds and their environments, along with the coordinated means, to eliminate the virus and maintain a negative herd. The tools described in this study will serve as a framework for regional and national PED eradication programs.

## 1. Introduction

Porcine epidemic diarrhea virus (PEDV), a highly infectious swine coronavirus, is the causative agent of porcine epidemic diarrhea (PED), which has a deadly effect on neonatal pigs in the first week of life, causing watery diarrhea, vomiting, fatal dehydration, and high mortality [1,2]. PEDV belongs to the genus *Alphacoronavirus* within the family *Coronaviridae* of the order *Nidovirales*. The virus possesses a single-stranded, positive-sense RNA genome of ~28 kb that encodes 16 nonstructural proteins (nsp1–16); 4 canonical coronaviral structural proteins, including the spike (S) glycoprotein; and a single accessory gene, ORF3 [3,4]. Based on the S sequence, PEDV strains can be phylogenetically divided into two genotypes with two subgenotypes: the low pathogenic (LP)-genotype 1 (classical G1a and recombinant G1b) and the highly pathogenic (HP)-genotype 2 (local epidemic G2a and global epidemic or pandemic G2b) [1,2]. Since the damaging pandemic that struck the American and Asian continents in 2013–2014, this swine enteric coronavirus has gained international attention, imposing economic and animal health burdens on the global swine industry [1].

PEDV was first described in South Korea three decades ago and it has persisted as an endemic pathogen, shifting from seasonal (winter) PED to a year-round disease and causing small- to large-scale outbreaks throughout the nation [1,2,5]. The most recent fatal epidemic of PED in South Korea occurred in late 2013 as a result of the arrival of the HP-G2b PEDV from the United States, where PED first emerged in April 2013 [6,7]. This PED calamity marked the advent of highly virulent strains of PEDV associated with massive neonatal mortality that led to the death of an estimated one million newborn piglets across the country, including Jeju Island, during 2013–2014 [6,7]. Since the 2013–2014 national disaster, the HP-G2b genotype of PEDV has been the dominant epidemic strain in South Korea [8,9,10,11].

Vaccination or intentional virus-exposure (feedback) practices have been implemented nationwide to manage PED by producing herd immunity, but PEDV has continued to impair swine health, leading to substantial financial losses within the domestic pork business. A large number of PED-affected swine farms have experienced recurrent outbreaks within one or two years, a situation which indicates that endemic or chronic infection of resident PEDV has become established on the farms, increasing the economic damage done to South Korea. When a disease becomes endemic, it is important to implement coordinated control measures (four pillars) to interrupt the chain of disease transmission at the farm level. These measures involve: (1) tightening biosecurity protocols and disinfection practices; (2) testing populations for serological and virus monitoring to check herd immunity levels in sows and to identify growing or finishing pigs with exposure to circulating PEDV; (3) vaccinating gilts and sows if necessary; and (4) improving herd management [1,2,12]. In the present study, we report a successful example of PED elimination in an endemically infected herd achieved by using the coordinated intervention approach, which is applicable to regional and national PED control and eradication.

## 2. Results

### 2.1. PEDV Monitoring and Surveillance Investigations

We first carried out nucleic acid and antibody-based monitoring and surveillance (MoS) of PEDV in a commercial farm affected by recurrent PED. All stool samples collected were positive for PEDV detected by rRT-PCR, and no other viral pathogens that cause diarrhea were detected. Next, the full S gene and the complete genome of the representative isolate with the lowest Ct value (15.6) were independently sequenced. The sequencing analysis revealed that the recurrent strain, designated as KOR/KNU-1907/2019 (referred to as KNU-1907), carried a 4161-bp segment of the S gene, predicted to encode a 1386-amino acid protein. It contained the genetic signature of the G2 field epidemic strains, including S insertions-deletions (S INDELs), compared with the prototype G1a CV777 strain [1,13]. As the isolate showed 95.5% amino acid homology with the Korean prototype G1b KNU-1406 and 99.2% homology with the Korean prototype G2b KNU-141112, it was classified as the HP-G2b subtype. Compared with KNU-141112, the recurrent KNU-1907 virus had 13 amino acid mutations in the S gene and formed its own genetic clade, as indicated by a barcode pattern illustrating each mutated locus (Figure 1). Furthermore, whole-genome sequencing showed that the KNU-1907 strain comprised 28,038 nucleotides, excluding the 3′ poly(A) tail, the size of the genome of most G2b field viruses, and had no INDELs, except for the S INDELs, in the entire genome. The KNU-1907 virus shared 99.5% homology with the KNU-141112 strain at the genomic level. The number of nucleotide and amino acid differences and the percent identities between KNU-1907 and KNU-141112 are summarized in Appendix A.

Based on genetic information about the S genes of the Korean PEDV strains collected in our laboratory, the barcode pattern of KNU-1907 is clearly distinguishable from that of the 2018–19 representative strains, KNU-1836, KNU-1904, and KNU-1911, which were identified in December 2018–March 2019 and were temporally and spatially similar to KNU-1907. However, KNU-1907 was analogous to the 2017 isolates, KNU-1704 and KNU-1708, collected in December 2017 that came from the same region where KNU-1907 was isolated (Figure 1). These data suggested that the KNU-1907 virus responsible for PED recurrence did not originate outside this farm. Rather, the virus might have remained in the farm since its first appearance in March 2018 and caused reinfection under altered field conditions after almost a year.

We also conducted serological examinations using VNT and ELISA to evaluate the immunity of sows against PEDV and to screen pigs with past exposure to the PEDV endemically circulating in the farm. All sows, including primiparous sows, developed PEDV-specific NAbs with titers ranging from 32 to 256 and anti-PEDV IgA antibodies, probably arising from natural or intentional infection or both (Figure 2, 3 November 2019). However, the kinetics of the NAb and IgA antibodies of sows, which should produce herd immunity, appeared to be variable (unstable) and of less sufficient levels for transferring adequate protective lactogenic immunity to offspring compared with those of PED-vaccinated sows [8]. Seroconversion was observed in growing (8 of 10) and finishing (10 of 10) pigs, indicating that the populations were exposed to PEDV in the wean-to-finish barns (Figure 3, 3 November 2019). Our data indicate that PEDV persisted within the farm, leading to endemic infection with PEDV that had been chronically circulating in the herd.

Subjective monitoring was performed to score the extent of adherence to biosecurity manuals using a checklist of “Do’s and Do not’s”, indicating practices that must or not be performed to reduce the risk of introduction (external biosecurity) and spread (internal biosecurity) of pathogens such as PEDV. The farm used in this research received the best score, 5, following on-site evaluation of the external and internal biosecurity practices at the yard, staff, and barn levels. However, despite full biosecurity management, the overall risk assessment indicated that this farm is considered to be at a medium risk of recurrence of PED due to relatively low sow immunity (score 3) and continual virus circulation (score 5) in the herd (Figure 4, middle pentagon).

### 2.2. Application of Two-Track Strategies for PED Eradication

The initial MoS data indicated that the enhancement of sow immunity and the severance of virus circulation were of primary importance. Therefore, we employed two-track strategies for PED elimination that simultaneously ameliorated the immunity of the pigs and eliminated the virus on this farm. The oral prime-parenteral boost prefarrow L/K/K scheme using new live and killed G2b vaccines was implemented to increase the capacity of protective immunity against PEDV. At the end of April 2019, two weeks after the completion of the first L/K/K vaccination, serum samples were collected from primiparous and multiparous sows and subjected to serology. The serological results confirmed the efficacy of the prime-boost vaccination on herd immunity, showing elevated and constant (stable) NAb and IgA antibody kinetics in sows compared with those in the previous month (Figure 2, 29 April 2019). However, 100% of the growing and fattening pigs tested were seropositive for PEDV, indicating continual spread of the virus within the farm (Figure 3, 29 April 2019).

As a first step toward implementing the second strategy regarding the removal of PEDV, the degree of virus contamination in the farrowing houses, the primary accommodation affected by acute PED, was assessed. To track PEDV, we collected fecal samples of sows and their suckling piglets from affected farrowing rooms during two months from March to May 2019 and performed rRT-PCR for detecting PEDV from individual rectal swabs. PEDV dynamics indicated that fecal shedding of the virus was gradually mitigated, and consequently, no animals tested on 13 May were found to shed PEDV in their feces (Figure 5a). Pen-side sampling of slurries began on April 29 to surveil the environmental stability of PEDV. PEDV was detected in four of six (67%) slurry samples from different farrowing rooms, although no PED-suspected symptoms occurred on the farm, suggesting heavy contamination with the virus in the herd and the environment (Figure 5b). Therefore, prompt execution of AIAO with thorough cleaning and disinfection protocols was recommended to minimize the potential sources of spreading the virus within the farm. Simultaneously, rRT-PCR-based PEDV MoS was carried out with pen-level slurry samples collected weekly for one month from 2 May to 3 June to monitor the outcome of the AIAO management. As shown by the mean Ct value kinetics, the concentrations of PEDV in individual slurries collected from different farrowing rooms lessened week by week, indicating that the AIAO and disinfection diminished the amount of the virus in the environment (Figure 5b).

### 2.3. Recurrent Infection in the Farrowing House on 18 June 2019

Despite attempts to control the virus, PED recurred on 18 June, mostly affecting 18- to 20-day-old piglets just before weaning, and was accompanied by moderate preweaning diarrhea without mortality. The S gene sequencing analysis showed that the virus responsible for this recurrence was nearly identical to the KNU-1907 strain identified in early March, with 99.6% homology and five amino acid changes. This genetic information indicated that PEDV continued to circulate in the farrowing houses. Subsequent serological tests indicated that sows developed high levels of PEDV-NAb and IgA antibodies in their colostrum and sera, yet the amount of maternal antibodies that their litters received waned in 20-day-old piglets (Figure 2, 26 June 2019). These data suggest that neonates within the first two weeks after birth, which retained enough of the lactogenic immunity gained from their dams, were likely resistant to the virus circulating in the farrowing rooms. In contrast, older piglets were vulnerable to virus infection because of a decrease in their maternal antibody titers. Serological screening implied that growing and finishing pigs were continuously exposed to the virus that persisted in the wean-to-finish barns (Figure 3, 26 June 2019). 

After the first reinfection in June 2019, the implementation of active, regular MoS of PEDV combined with disinfection was extended to the entire pig housing, including the farrowing rooms, nursery barns, and finisher barns, to attain virus elimination in this farm, while continuing to employ and monitor the routine prime-boost prefarrow L/K/K scheme to sustain the extent of protective immunity in piglets until weaning. In July 2019, we carried out serological monitoring to assess herd immunity levels in piglets of different ages from primiparous or multiparous sows. Although the amounts of maternal NAbs were higher in one-week-old piglets than three-week-old piglets and in piglets from multiparous sows than in those from primiparous sows, the protective capacity of the maternal immunity was well maintained during the suckling period (the first three weeks after birth) (Figure 6). These results showed that the prime-boost prefarrow vaccination provided an improvement in the extent and stability of herd immunity.

Pen-side slurry samples collected weekly or monthly from 20 farrowing rooms, 24 nursery rooms, and 8 finisher barns were subjected to rRT-PCR. The MoS conducted on 26 June found that seven farrowing rooms (35%) were severely contaminated with PEDV, the S gene sequence of which was 100% identical to that of the virus strain collected on 18 June (Figure 7). However, the number of contaminated barns decreased thereafter, as a result of the AIAO and continuous disinfection. We were able to certify that all farrowing rooms in three houses were virus free on 5 August 2019, and since then, the virus-free status has been preserved until the time of writing (Figure 7). Consistent with the serology data from growing and finishing pigs, the 26 June MoS revealed that numerous nursery rooms throughout three barns were contaminated with PEDV, which were more serious compared with the infection in the farrowing houses (Figure 8). Four of eight finishing barns tested were also PEDV positive (Figure 9).

### 2.4. First Potential Recurrence in the Nursery Barn in November 2019

While reinforcing the internal biosecurity and disinfection practices to prevent the spread of the virus within the nursery barns as well as to the farrowing houses and to clean the virus from wean-to-finish barns, MoS involving molecular and serological screening was continued. The slurry-based rRT-PCR revealed that until the middle of September, 30–50% of the nursery rooms were positive for PEDV, although the number repeatedly decreased and increased (Figure 8). The serology data collected on 22 August indicated that the growing and finishing pigs had been exposed to circulating PEDV (Figure 3, 22 August 2019). On 23 September 2019, the number of contaminated nursery rooms was dramatically reduced, and two rooms remained PEDV positive, suggesting the probability of virus elimination. 

Despite retaining PEDV-free status in the farrowing houses, the number of PEDV-positive nursery rooms (13 of 24) alarmingly increased again on 11 November (Figure 8), and four finisher barns were recontaminated with PEDV (Figure 9). The S sequencing analysis demonstrated that the virus detected in November (KNU-1907-6) was virtually identical, at 99.5–99.9% homology, with seven amino acid changes from the original March (KNU-1907) strain and two amino acid changes from the June (KNU-1907-3) strain (Figure 10), indicating that PEDV continued to spread within the farm. The subsequent seroprevalence study of grow/finish pigs corroborated the results of molecular and genetic diagnostics (Figure 3, 9 December 2019). Within one month, PEDV was detected in all nursery rooms, heralding the possibility of PED recurrence five months after the June recurrence. Thus, active surveillance on the farm was strengthened, and serological assays were conducted to check whether the level of immunity of sows was sufficient for conferring protective immunity to their litters via lactation. The serology results confirmed that very high levels of PEDV-Nabs, and IgA antibodies were maintained in the colostrum of primiparous sows, as well as in milk collected at one and two weeks postfarrowing (Figure 2, 25 November 2019).

On 3 December, a suspected infection of PED appeared simultaneously in several replacement gilts that were purchased from external conventional pig breeding herds and had been acclimated in finisher barns. The affected gilts started showing mild diarrhea and tested positive for PEDV. Genetic analysis revealed that the virus (KNU-1907-8) that infected the gilts was 100% equal to that collected in the nursery barns on 25 November (KNU-1907-7) (Figure 10), suggesting that the gilts were exposed to circulating PEDV during an acclimation period. This reinfection case in replacement gilt populations was terminated without secondary transmission to other herds, particularly pregnant and farrowing facilities, due to active surveillance and early response.

### 2.5. Second Potential Recurrence in the Nursery Barn in April 2020

The challenge of eliminating the virus continued through the disinfecting and remodeling of the nursery barns, followed by the implementation of active MoS. As a result, the number of PEDV-contaminated rooms was significantly reduced by 10 February 2020, even though four rooms remained positive for PEDV. This reduction was transient, and all nursery rooms, except for two, were recontaminated with PEDV on 29 April, again posing a potential risk of PED recurrence on this farm (Figure 8). However, despite the continuation of PEDV circulation within the nursery and finisher barns, no case of suspected PED occurred on the farm. Serological screening of low-parity sows (parities 1–2) conducted on 20 May confirmed the maintenance of ample and stable quantities of colostral NAbs and IgA antibodies, indicating the establishment of herd protective immunity (Figure 3, 15 May 2020).

### 2.6. Eradication of PEDV from This Farm

The decontamination of nursery barns began in July, and all wean-to-finish barns tested negative for PEDV on 21 October 2020 (Figure 8 and Figure 9). Since then, the PEDV-negative condition has been maintained in the nursery and finisher barns until the time of writing. To verify the virus-free status, 40 serum samples were collected three times from growing (*n* = 20) and finishing (*n* = 20) pigs in December 2020 through February 2021 and subjected to VNT. No animals seroconverted against PEDV, suggesting the absence of virus circulation (Figure 3, 16 December 2020, 14 January 2021, 15 February 2021). The MoS data accumulated in this study indicated that the PED-infected farm recovered its PEDV-free status in two years, thereby indicating successful virus eradication. As expected, the final risk assessment indicated that this farm is now no longer at risk of PED recurrence (Figure 4, right pentagon).

### 2.7. Genetic Characteristics and Evolution of PEDV in the Farm

Twelve full S sequences, including three complete genome sequences, of PEDV (KNU-1907) were determined in this farm during this study (Table 1). We were able to sequence the full genome of PEDV in fecal samples collected from diarrheic pigs but not those in pen-level slurries, because viral concentrations in PEDV-positive slurry samples were not high enough for whole-genome sequencing. The S sequences were compared among PEDV isolates, according to their collection dates (Figure 10). A total of 5–14 amino acid substitutions were randomly distributed in PEDVs that were identified chronologically. Among them, three changes, including N29K, F604Y, and E608D, were evident in NTD/S0 (residues 19–220) and “collagenase equivalent” (COE) (residues 502–641) neutralizing epitopes in HP-G2b PEDV [14,15]. These S mutations occurred simultaneously in the isolates collected on 20 May 2020 (KNU-1907-10), 15 months after the first sequenced PEDV in the farm. The S sequences obtained were further compared with reference sequences of global PEDV strains, including all four genotypes (Figure 11). 

The whole-genome sequences of PEDV isolates that were obtained in March, June, and December were analyzed (Figure 12). Compared with the March isolate, the June (KNU-1907-3) isolate contained 23 nucleotide and 16 amino acid changes, and the December (KNU-1907-8) isolate contained 51 and 23 substitutions at the nucleotide and amino acid levels, respectively. Although nucleotide substitutions were observed throughout the genome, amino acid mutations commonly accumulated in nsp2, nsp3, nsp4, nsp5, nsp12, nsp15, S, ORF3, and the nucleocapsid of the June and December isolates; the latter had one additional substitution at each terminus of the genome. The number of nucleotide and amino acid differences and the percent identities among KNU-1907 isolates are summarized in Appendix A. The PEDV S sequences were analyzed using the BEAST 2.6.3 package. The substitution rates were estimated as 1.683 × 10^−4^ substitutions/site/year and 2.239 × 10^−3^ substitutions/site/year at nucleotide and amino acid levels, respectively (Table 2). The evolutionary rate estimated for the whole genome was 4.921 × 10^−7^ substitutions/site/year.

## 3. Discussion

PED management is not very practicable in swine production systems, especially in farrow-to-finish enterprises, because there still are gaps in our knowledge about the control measures that break the cycle of transmission at the farm and industry levels. The absence of active intervention programs has helped exacerbate PED, leading to it being endemic in most Asian pig-raising countries, including South Korea. Since the 2013–2014 disastrous epidemic in South Korea, seasonal PED has become a year-round disease, and recurrent outbreaks in PED-affected herds have occurred frequently across the nation. The endemic nature of the disease has made PED control even more difficult, and its impacts have become more significant. Moreover, eliminating PED from endemic farrow-to-finish farms is a complicated task because of the continue flow nature of the animals at the farm and the level of environmental contamination. Therefore, it is recommended that pig producers apply the four pillars of control, which cannot be ignored, to successfully manage PED in endemically infected herds. These four pillars are: (1) biosecurity, (2) diagnostics and MoS, (3) vaccination, and (4) herd management [1,2,12]. In the current study, we aimed to systemically apply the coordinated control measures and monitor biosecurity practice, herd immunity, and virus circulation to manage and eradicate PEDV in a PED-infected farm.

We first conducted PEDV diagnostic testing using RT-PCR and nucleotide sequencing to obtain timely genetic information about the virus, which is of primary importance in recurrent outbreaks. The genetic details of PEDV collected at the time of recurrence on the farm indicated that the virus might have persisted since the first occurrence. Next, we executed an assessment model system that evaluates the status of biosecurity, herd immunity, and virus circulation through on-site data collection and laboratory examination to estimate the risk of PED recurrence in the farm. Although this farm followed all external and internal biosecurity procedures strictly, it was at risk of a recurrence of PEDV. This susceptibility resulted from low herd immunity and ongoing virus circulation on the farm, as shown by the serological evidence in this study. The first evidence was obtained using VNT and IgA ELISA to measure the immunity of sows, which pass on lactogenic maternal antibodies to suckling piglets for protection against PEDV. The second line of investigation used only VNT to detect PEDV-seropositive pigs that had been exposed to the virus circulating in wean-to-finish barns. Therefore, the serology must be performed bidirectionally in sows and their offspring from farrowing houses as well as in grow/finish pigs from the nursery and finisher barns to obtain valuable information from PED-affected farms. Our data also suggested that adopting a feedback practice in this farm at the emergence of the disease might succeed in minimizing the time period of neonatal mortality but appeared to fail at assuring immunity and the cessation of shedding of PEDV within the population, thereby leading to recurrence within a year. Given the drawbacks of a long-term view, feedback that may help mitigate PED and induce immunity through controlled oral exposure should be contemplated only in urgent explosive epidemic cases with very high mortality and morbidity.

Based on our initial MoS information, a two-track control strategy was introduced in this farm to elevate the immunity of sows and to break the chain of virus circulation. This strategy was evaluated using longitudinal laboratory tests to monitor its effects. Colostral antibodies received from immune dams enable their offspring to combat PED. Sow immunity is therefore a vital parameter required for the prevention and control of PED [2]. In South Korea, multiple-dose prime-boost PEDV vaccination programs prior to farrowing or breeding have been commonly advised for decades for the application of gilts and pregnant sows [1]. Growing evidence emphasizes the oral route of prime immunization in terms of the effectual priming of the gut using a live vaccine or through feedback to augment and sustain lactogenic immunity against PEDV [8,16,17]. Using the lessons gained from previous research, we developed a safe and effective new orally administrated HP-G2b live vaccine that is now available on the domestic market [8]. Although the unavailability of an efficient oral vaccine for HP-G2b PEDV was one of the missing pieces of the four-pillar-based control measures until recently, the launch of an oral vaccine allowed us to complete the four pillars. In this study, we successfully applied the new oral vaccine for the prime-boost prefarrow L/K/K regimen with the parenteral killed commercial G2b vaccine. This vaccination scheme significantly improved the NAb and IgA antibody kinetics in sows, establishing and maintaining herd immunity.

The second strategy focused on virus screening in feces from individual pigs and pen slurries from the environment. rRT-PCR diagnostics can detect PEDV in rectal swabs of infected pigs for up to two months after infection [18,19]. PEDV was detectable in stool samples from piglets until the end of April, when two months had passed since the PED recurrence in late February. As PEDV was still found in the herd and the environment at that time, AIAO management in parallel with aggressive cleaning, disinfection, and sanitation protocols began to sever virus circulation. As a result, no PEDV-positive pigs in the farrowing houses were diagnosed using rRT-PCR in the middle of May. However, notwithstanding the cessation of PED, slurries from different farrowing rooms tested positive for PEDV, although the amount of positivity was reduced, likely owing to the influence of AIAO and disinfection practices. According to experimental bioassays, PEDV survives for more than 7 days in inoculated fresh feces, whereas the virus can survive for ≥14 days at RT and >28 days in a slurry stored at −20 °C–4 °C, regardless of relative humidity levels. These findings demonstrate the greater survivability of PEDV in infected manure than in feces [20]. The experimental observations in a previous study were terminated after 28 days [20]; therefore, the longest duration that PEDV can survive and be infectious in the environment could be more than that observed under experimental conditions. In this study, we observed the detection of PEDV in the infected slurry over long periods under field conditions, indicating that slurry samples could be helpful to demonstrate the persistence and even replicability of PEDV in the environment. Another study revealed that PEDV could survive up to nine months in on-farm manure storage in infected swine farms under temperatures ranging from −30 °C to 23 °C [21]. The slurry was superior to feces for detecting PEDV in the environment. Therefore, sampling of slurries and monitoring of PEDV in the slurries have continued, following AIAO and disinfection practices.

PEDV recurred in June and genetic data collected at this time demonstrated that PEDV infection spread in the farrowing houses. However, unlike the first recurrence, PEDV reinfection had little effect on newborn piglets but caused preweaning scours in piglets aged ≥ 18 days, with no mortality in the affected pigs, although the severity and mortality of PEDV is age dependent and occurs almost exclusively in neonatal piglets less than 1 week old [1]. On-site inspection and serological evidence showed that the intake of colostrum and milk by newborn piglets was well managed on this farm, despite the high levels of immunity of the sows, but maternal antibodies subsided in 20-day-old piglets. Given multiple challenges, including virus contamination in farrowing houses and decreased maternal immunity in older piglets, preweaning diarrhea was predictable in these populations. In this scenario, passive lactogenic immunity acquired from vaccinated sows can protect litters against PEDV when viral loads in the environment do not exceed the protective capacity of maternal antibodies. If lactogenic immunity in piglets becomes depleted or decreased due to illness, atrophy, inadequate colostrum intake, or age dependency, PEDV can infect immunocompromised susceptible piglets that will multiply and shed large amounts of viruses in their feces, thereby contaminating the environment. This contamination may lead to virus transmission to other piglets, causing diarrhea, if viral loads in the environment surpass the protective levels of their lactogenic immunity. If a high level of immunity was present in sows in this farm, they conferred ample amounts of passive milk antibodies to sucking piglets. In this case, younger neonates who maintain the protective capacity will be protected against PEDV, irrespective of the viral dose in the environment, whereas older neonates who lose their protective maternal antibodies will become infected. Nevertheless, if there are no breaks in a vicious contamination-transmission-infection cycle of PEDV in the environment, the virus will be able to re-enter the farrowing houses, resulting in diarrhea and deaths of newborn piglets when they lack protective immunity. As PEDV is a highly contagious virus with a low infective dose [22], the virus-infected slurry can readily contaminate the uninfected environment, leading to significant circulation of the virus in the herds. Therefore, we extended slurry sampling and longitudinal monitoring of PEDV detection in the environment to the whole yard to avoid missing evidence of the virus.

At five months postrecurrence on this farm, PEDV was eliminated from the farrowing houses in August 2019 as a result of coordinated active monitoring and management efforts. However, viral and serological screening indicated that there was survival and spread of PEDV in wean-to-finish barns. In particular, early rRT-PCR-based MoS on 26 June detected virus circulation throughout nearly all nursery barns. As environmental monitoring of PEDV in wean-to-finish barns first began at this time, we cannot exclude the possibility that there had been environmental contamination by PEDV since the recurrence. However, we suspected another factor, such as the management of infected piglets. In the PEDV reinfection on 18 June, early weaning was performed to remove piglets that became infected. It was anticipated that the movement of infected piglets to weaning pens would be the main source of magnifying the extent of virus contamination in nursery barns. Thus, it is recommended for pig farmers to employ depopulation by euthanizing affected piglets to avoid disease transmission to other barns within the farm instead of early weaning in enzootically infected farms.

Cooperative efforts to decontaminate virus circulation resulted in a decrease in PEDV from the nursey barns until the end of September 2019, when only two barns remained contaminated. Virus circulation was reproduced in November, caused by unknown factors, which suddenly increased the number of PEDV-contaminated barns from November to December 2019, potentially threatening the on-farm health status. During this period, active MoS was strengthened to monitor all herds closely and to quickly cope with the situation when pigs that showed clinical signs of PEDV were observed. At the end of November, we found purchased replacement gilts, which became infected with PEDV during acclimation in finisher barns. However, the incidence of the virus was limited to the gilts, and further disease transmission did not occur. Therefore, this risk category was named the first potential recurrence but did not lead to an actual outbreak of PED. In this instance, the virus might be hidden in the environment of the nursery barns that were PEDV-positive according to the last monitoring and may play a role in a “Phantom Menace (PM)” that can stealthily infect pigs, amplify, contaminate, and spread to other pigs. As the subclinical spread of PEDV occurs in grow/finish pigs, with the animals developing a humoral immune response against the virus, it is necessary to monitor swine populations in affected farms using complementary virus and serological screening to identify the PM virus itself or the pigs with exposure to the virus in their environment and to further intervene in the chain of virus circulation.

PEDV infection of gilts indicates that virus contamination in wean-to-finish barns must be removed, or the farm faces a risk of recurrence. It is generally acknowledged that PEDV infection is self-limiting or asymptomatic in weaner to finisher pigs; however, the virus is shed in their feces, contaminates their barns, and replicates in a subclinical manner in these populations [2]. Thus, PEDV can persist asymptomatically in grow/finish pigs in endemically infected farms through the contamination-transmission-infection cycle, augmenting opportunities for gilts to encounter the virus during acclimation in contaminated barns. Although purchased replacement gilts pose a greater health risk to the breeding herd than internally reared gilts, both gilts will indifferently confront a risk of PEDV infection in endemically affected farms. Once infected, asymptomatic gilts can serve as “Trojan Pigs” that present no clinical signs but produce and shed viruses, which will be transferred to the farrowing house when the gilts are used as replacements, carrying viruses in their feces. In this way, viruses shed by the “Trojan Pigs”, even at undetectable viral loads, can spread to and replicate in neonates, thereby acting as the source of acute PED recurrence that affects piglets with low or no passive immunity in a clinical manner. Thus, gilt management, particularly purchased gilts, is pivotal to the control of PED in affected farms.

Although a second fade-out of PEDV from the contaminated nursery barns was observed in February 2020, virus contamination was transmitted throughout most nursery rooms from April to May 2020. However, no typical PEDV infection was reported on this farm during this period. The monitoring data collected in December 2019 indicated that the nursery barns were seriously contaminated, whereas the finisher barns tested as virus free. The data obtained in February 2020 show that the nursery barns were largely decontaminated, whereas the finisher barns were significantly recontaminated (Figure 8 and Figure 9). These results suggested a revival of the “PM” virus that had existed on the farm, which reproduced again in the nursery room around December 2019, spread to the finisher barn before February 2020, and re-entered the nursery barn thereafter, probably through contaminated sources, including pigs and humans. These data indicate how PEDV moves in a farm with an endemic infection and suggest that we should be wary of virus circulation in the environment, herd, or both, which could significantly hinder PED eradication. The two recurrence scares during this study had no impact on swine populations, and this outcome may account for the high-level sustainment of sow immunity and biosecurity that otherwise could have suppressed the potential outbreak caused by virus circulation on the farm with endemic disease. The last PEDV detection was in wean-to-finish barns in August 2020, and since then, the PEDV-free status has been recovered and maintained on this farm until the end of the study (February 2021). All the important events during the PED elimination project are summarized in Figure 13.

The full S sequences of PEDVs were temporally collected and further analyzed to determine the evolution rate of KNU-1907. Previous studies estimated that the substitution rates for the PEDV G2b mainland and Jeju Island strains are 7.18 × 10^−4^ substitutions/site/year and 14.80 × 10^−4^ substitutions/site/year in the S gene, respectively, indicating that genetic evolution proceeds at a faster rate on Jeju Island than on the South Korean mainland [10,11]. However, the substitution rates of PEDV in our present study farm were 1.683 × 10^−4^ substitutions/site/year in the S gene and 4.921 × 10^−7^ substitutions/site/year in the whole genome, which is lower than the results reported by previous studies [10,11]. While previous reports of PEDV evolution were based on the viruses from different farms in the mainland and Jeju Island, the present study analyzed virus infections in a single farm. The data suggested that PEDV on a swine farm undergoes slower evolution than the overall PEDV evolution rate because farm-to-farm transmission barriers are excluded. Despite the slow evolution of PEDV, sequence analysis highlighted the emergence of 14 amino acid mutations in S and 23 in the full genome. In particular, three amino acid substitutions arose in two NTD/SO and COE neutralizing epitope regions in PEDV [14,15]. The development of the mutations might be temporally identical because they all were found concurrently in slurry samples collected on 20 May 2020, suggesting a potential for the antigenic change. We further investigated the putative antigenicity of the S gene using the Bepipred linear epitope prediction method [23]. The predicted epitopes showed a similarity to a potential B-cell epitope, corresponding the 19–220 and 502–641 epitope regions, between the primary (KNU-1907) and later (KNU-1907-10) strains, indicating the absence of antigenic drift despite the amino acid mutations that occurred throughout 15 months (Appendix A).

## 4. Materials and Methods

### 4.1. Farm Information

A 1500-sow commercial farrow-to-finish farm was selected for this study. The farm is located in the Gyeongbuk Province in southeastern South Korea and has no neighboring swine farms within 5 km. The farm contains 17 main buildings, including one breeding barn; two gestation barns; three farrowing houses with separate rooms 1–8, 9–16, and 17–20; three nursery barns with separate rooms 1–11, 12–19, and 20–24; and eight finisher barns (Appendix A).

In March 2018, PEDV first appeared in this farm, causing acute PED infection with lethal watery diarrhea and >90% mortality in newborn piglets. To control PED, sows were intentionally exposed to PEDV via oral exposure in the form of homogenized intestines from infected pigs in the early phase of the outbreak. This practice was used to initiate and maintain herd immunity and it could gradually mitigate the significant losses on this farm. Since then, no further PED outbreaks occurred on the farm until early 2019. At the end of February 2019, PED recurred on this farm, primarily affecting first-parity sows and their offspring and producing 50–60% mortality. Fecal samples collected from diarrheic pigs were submitted to our laboratory for diagnosis and further management.

### 4.2. Sample Collection

Collection of individual pig samples (feces, sera, and/or colostrum) and pen-side samples (slurry) was started on 11 March 2019, two weeks after the PED recurrence in the farm, and was continued until February 2021. Stool samples from diarrheic pigs were collected using 16-inch cotton-tipped swabs. Pig slurry specimens were obtained over time using syringes from slurry pits in 20 farrowing rooms, 24 nursery rooms, and 8 finisher barns. Fecal and slurry samples were diluted with phosphate-buffered saline (PBS) to produce 10% (wt/vol) suspensions. The suspensions were vortexed and centrifuged for 10 min at 4500× *g* (Hanil Centrifuge FLETA5, Incheon, Korea). The clarified supernatants were tested by PEDV-specific real-time RT-PCR and were further subjected to nucleotide sequence analysis, if necessary. Blood was periodically taken from first-parity (primiparous) and second-parity or greater (multiparous) sows and pigs of different ages, as described in the legends to Figure 2, Figure 3, and Figure 6. Colostrum and/or milk were collected on or after the day of farrowing from primiparous and multiparous sows, as mentioned in the legend to Figure 2. Serum and colostrum samples were centrifuged and tested using a virus neutralization assay or enzyme-linked immunosorbent assay (ELISA) to characterize the antibody kinetics.

### 4.3. Quantitative Real-Time RT-PCR (rRT-PCR)

Viral RNA was extracted from fecal samples prepared as described earlier, using i-TGE/PED Detection Kits (iNtRON Biotechnology, Seongnam, Korea) according to the manufacturer’s protocol. PEDV S gene-based quantitative rRT-PCR was performed using One-Step SYBR PrimeScript RT-PCR Kits (TaKaRa, Otsu, Japan) with the forward primer 5′-ACGTCCCTTTACTTTCAATTCACA-3′, reverse primer 5′-TATACTTGGTACACACATCCAGAGTCA-3′, and a probe 5′-FAM-TGAGTTGATTACTGGCACGCCTAAACCAC-BHQ1-3′, as previously described [11,24,25,26]. The reaction was performed using a Thermal Cycler Dice Real-Time System (TaKaRa) according to the manufacturer’s protocol, under the following conditions: 1 cycle of 45 °C for 30 min, 1 cycle of 95 °C for 10 min, and 40 cycles of 95 °C for 15 s and 60 °C for 1 min. The results were analyzed using an automatic baseline as described previously [25,26]. Samples with mean cycle threshold [Ct] values of <35 were considered to be positive for PEDV.

### 4.4. Nucleotide Sequence Analysis

The S glycoprotein gene sequences of PEDV-positive samples were determined using traditional Sanger methods. Two overlapping cDNA fragments spanning the entire S gene of each isolate were amplified by RT-PCR, as previously described [13]. The individual cDNA amplicons were gel-purified, cloned using the pGEM-T Easy Vector System (Promega, Madison, WI, USA), and sequenced in both directions using two commercial vector-specific T7 and SP6 primers and gene-specific primers. The complete genomes of representative PEDV isolates with mean Ct values of <20 were also sequenced. Ten overlapping cDNA fragments spanning the entire genome of each virus strain were amplified by RT-PCR as described previously [7,10,11,25,27], and each PCR product was sequenced as described above. The 5′ and 3′ ends of the genomes of the individual isolates were determined by rapid amplification of cDNA ends, as described previously [28]. The full S gene or whole-genome sequences of PEDV identified on the farm have been deposited in the GenBank database under the accession numbers MW560720–8 and MW560715–7, respectively.

The sequences of fully sequenced S genes of global PEDV isolates were aligned using the ClustalX 2.0 program [29], and the percentages of amino acid sequence divergences were assessed using the same software. Phylogenetic trees were constructed from an alignment of the amino acid sequences using the neighbor-joining method and were subjected to bootstrap analysis with 1000 replicates to determine the percentage reliability values for each internal node of the tree [30]. A phylogenetic tree was generated using the Mega X software [31]. 

Bayesian analyses were performed with BEAST v2.6.3 to estimate the rate of evolution of the full S and whole-genome sequences [32]. BEAST priors were introduced with BEAUTi v2.6., including a relaxed clock log model with a lognormal rate distribution and an exponential growth coalescent model of population size and growth. A Markov chain Monte Carlo (MCMC) algorithm was run using 20 million states and sampling every 400 steps. The convergence of the MCMC chains was checked using Tracer v.1.7.1, ensuring that the effective sample size values were ≥200 for each estimated parameter.

### 4.5. Virus Neutralization

The presence of PEDV-specific neutralizing antibodies (NAbs) in serum and colostrum samples collected from the pigs was determined using a conventional virus neutralization test (VNT) in 96-well microtiter plates with PEDV isolate KNU-141112 as previously described [27,33], with minor modifications. Vero cells at 2 × 10^4^ cells/well were grown in 96-well tissue culture plates for 24 h. KNU-141112-P5 virus stock was diluted in serum-free α-MEM to achieve 200 TCID_50_ in a 50 μL volume. Next, the diluted virus was mixed with 50 μL of 2-fold serially diluted (1:2 to 1:512) inactivated serum or colostrum samples and incubated at 37 °C for 1 h. The mixture was added to Vero cells and incubated at 37 °C for 1 h. After removing the mixture, the cells were rinsed with PBS five times and maintained in a virus growth medium [10,25] at 37 °C in a 5% CO_2_ incubator for 2 days. The neutralizing endpoint titers were calculated as the reciprocal of the highest serum dilution that inhibited the virus-specific cytopathic effects by ≥80% relative to the controls in duplicate wells. The serum samples with neutralizing endpoint titers of ≥1:4 were considered to be positive for the PEDV-neutralizing antibody.

### 4.6. Enzyme-Linked Immunosorbent Assay

Recombinant PEDV S1 protein was purified from PK-rS1-Ig cells as described previously [33]. Anti-PEDV immunoglobulin A (IgA) antibodies in colostrum or milk samples were detected using an *in-house* PEDV G2b S1-based indirect ELISA as described previously [8,34,35], with minor modifications. Microtiter plates (Nunc, Naperville, IL, USA) were coated with 0.5 ng of the S1 antigen diluted in 50 mM bicarbonate coating buffer (pH 9.6) and incubated overnight at 4 °C. After three washes with PBS containing 0.05% Tween 20 (PBST), the plates were blocked with 5% powdered skim milk (BD Biosciences, Belford, MA, USA) in PBST for 2 h at 37 °C and then incubated with each serum sample diluted 1:100 in PBST containing 10% goat serum (Vector Laboratories, Burlingame, CA, USA) for 1 h at 37 °C. After washing, a 1:20,000 diluted peroxidase-conjugated goat anti-porcine IgA (Abcam, Cambridge, UK) was added to each well and incubated at 37 °C for 1 h. The peroxidase reaction was visualized using tetramethylbenzidine-hydrogen peroxide as the substrate (R&D Systems, Minneapolis, MN, USA) for 20 min at room temperature (RT) in the dark and was stopped by adding 2N sulfuric acid (R&D Systems) to each well. The optical density (OD) of each sample was measured at 450 nm using a SPARK 10M multimode microplate reader (TECAN, Männedorf, Switzerland). Positive control, negative control, and blank (sterile water) samples were included in each plate; all of the clinical and control samples were tested in duplicate.

### 4.7. Biosecurity Monitoring

Subjective monitoring encompassed the classic approach based on the data collected from farms and surveys and verification of the checklists of established tasks. External and internal biosecurity protocols were reviewed by a herd veterinarian using on-site inspection and in-person interviews and were evaluated at the yard, staff, and barn levels using purpose-designed checklists with a 1–5 scoring scale, wherein 1 = worst and 5 = best, which is based on the PEDV Biosecurity Quick Fact Sheet (https://www.manitobapork.com/images/producers/pdfs/biosecurity/PEDv-Biosecurity-Quick-Facts.pdf, accessed on 16 January 2021), with some modifications to adjust to the circumstances of domestic pig farming, as described previously [36].

### 4.8. Risk Assessment with the Pentagon Profile System

A pentagon profile system, used to assess the risk of PED recurrence, was created based on the evaluation scores independently acquired from the biosecurity-related questionnaire (1 = worst; 5 = best), the measurement level of sow immunity (1 = worst; 5 = best), and the endemic status (0 = best; 5 = worst), as described previously [36]. The subjective monitoring scored how well each swine producer followed all biosecurity protocols at the yard, staff, and barn levels, as described above. The level of herd immunity was estimated based on the stability and degree of NAbs and IgA antibodies against PEDV in sows. The endemic status of the infection was determined according to the circulation of PEDV in the herd, seroconversion against PEDV in growing pigs, or both. Individual scores from the five factors were used to construct the five arms of a risk pentagon.

### 4.9. PED Vaccination Program

Live prime-killed/killed boost (L/K/K) multiple-dose PED vaccination at two-week intervals starting prior to farrowing has been implemented using commercial second-generation vaccines since March 2019. All sows, including gilts, were orally administered a G2b live oral vaccine (KNU -141112 S-DEL5/ORF3 strain, PED-X Live^®^, CAVAC, Daejeon, Korea) once, using drenching in accordance with the manufacturer’s instructions, and were then intramuscularly (IM) boosted twice with a commercial inactivated G2b PEDV vaccine (ISU46065IA13 strain, PED-X^®^, CAVAC) at 6, 4, and 2 weeks prepartum or prebreeding, respectively.

### 4.10. All-in-all-out (AIAO) Management

A modified version of AIAO management was implemented using spare virus-free farrowing rooms in May 2019. This procedure involves moving all sows from virus-contaminated rooms to extra rooms; cleaning, disinfecting, and drying the contaminated rooms; and returning the sows to the cleaned rooms. The disinfection protocols used to sanitize swine facilities that have housed PEDV-positive animals were performed in the following sequence, repeated three times, as described previously [1,2,12]: (1) cleaning using a high-pressure washer and warm water at temperature ≥70 °C; (2) disinfection with a suitable disinfectant according to the directions on the label; and (3) overnight drying.

### 4.11. Statistical Analysis

All values are expressed as mean ± standard deviation of mean (SDM). Statistical analyses were conducted using Student’s *t*-test. *p* values of less than 0.05 were considered to be statistically significant.

## 5. Conclusions

The present study is the first to report a case of PEDV elimination in an endemically infected farrow-to-finish pig farm. To accomplish the final goal of the eradication of PEDV, conducting a risk assessment of PED recurrence is critical for the timely gathering of information about pathogen exposure and immune responses in PED-affected farms. This assessment involves: (1) molecular and genetic diagnostics (virus screening) of PEDV to obtain information on whether the virus has remained inside the farm after the outbreak or was newly introduced from outside the farm; (2) serological screening in farrowing herds to obtain information about sow protective immunity against PEDV; (3) viral and serological screening in growing and finishing herds to obtain information on virus circulation within the farm; and (4) subjective biosecurity monitoring to obtain information on how well the farmer follows biosecurity protocols. According to the results of the risk assessment, the four-pillar-based coordinated intervention measures are actively carried out to enhance the immunity of the hosts and diminish the infectivity of PEDV in the farrow-to-finish farm suffering from endemic virus. These measures include: (1) strict internal and external biosecurity; (2) prime-boost prefarrow L/K/K vaccination and longitudinal monitoring of protective immunity in sows and their offspring; (3) secure cleaning/disinfection practices, combined with the AIAO management, if needed, in farrow houses and longitudinal MoS of PEDV in the herd and the environment (slurry); and (4) disinfection and gilt management in wean-to-finish barns, in parallel with longitudinal MoS of PEDV in the herd and the environment (slurry). Figure 14 diagrams a decision tree to explain the rationale of elimination strategies and measures following PEDV exposure in farrow-to-finish farms. Our study demonstrated that PEDV elimination was difficult in wean-to-finish barns that were once contaminated and served as incubators of virus circulation and reinfection, and it also illustrated the contribution of the vaccination and biosecurity regarding the prevention of PED recurrence. In particular, low-dose infections and high viability in the environment were major challenges in the eradication of PED in this study. Thus, we need to trace PEDV in the environment by regularly collecting and testing slurry samples from farrowing houses and wean-to-finish barns because PEDV survives well and maintains infectivity in slurry. In addition, since slurry can reflect the risk of manure spreading, we should manage manure storage manuals and treatments to reduce the load and infectivity of PEDV to decrease the risk of recurrent infection within a farm and the further spread of viruses to nearby farms and even further afield. Our next goal is defined as establishing regional control programs targeted at PED by systematically expanding the application of the aforementioned PED intervention tools to swine farm-dense regions across the country. The current study underscores the importance of time-consuming control measures and cooperative efforts to eliminate PEDV in farms with endemic infections. Our collaborative on-site and laboratory-based tools for breaking the virus circulation cycle at the farm level provide a cornerstone upon which to found national PED control and eradication programs.

## Figures and Tables

**Figure 1 pathogens-10-00830-f001:**
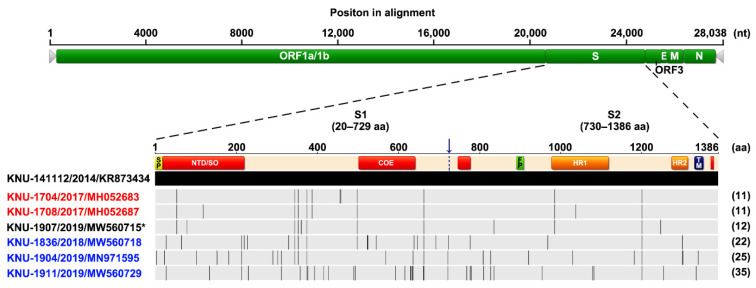
Schematic diagram of multiple alignments of the PEDV full S gene relative to the Korean prototype G2b strain KNU-141112 using Geneious software version 10.2.4. The top illustration represents the genomic regions, with green bars symbolizing the identified open reading frames (ORFs). Light gray arrows represent each 5′- and 3′-untranslated region. The second diagram displays the organization of the S protein, indicating the S1 and S2 domains that contain a signal peptide (SP), an N-terminal hypervariable domain (NTD), a fusion peptide (FP), heptad repeat regions (HR1 and HR2), and a transmembrane domain (TM). Red areas in the diagram depicting the S protein represent four neutralizing epitopes (NTD/SO, residues 19–220; COE, residues 502–641; residues 744–774; residues 1371–1377) of HP-G2b PEDV. The 2017–19 PEDV strains that were identified in our laboratory are color coded and include KNU-1704 and KNU-1708 (red) identified in December 2017 and KNU-1836, KNU-1904, and KNU-1911 (blue) identified in December 2018–March 2019. The KNU-1907 strain (black) identified in this study is marked with an asterisk (*). Lightly shaded areas are those identical to KNU-141112, and the vertical black bars represent one amino acid sequence that is divergent from that of KNU-141112. The strain name, isolation year, and GenBank accession number are shown on the left. The digits in parentheses on the right indicate the number of amino acid changes compared with KNU-141112.

**Figure 2 pathogens-10-00830-f002:**
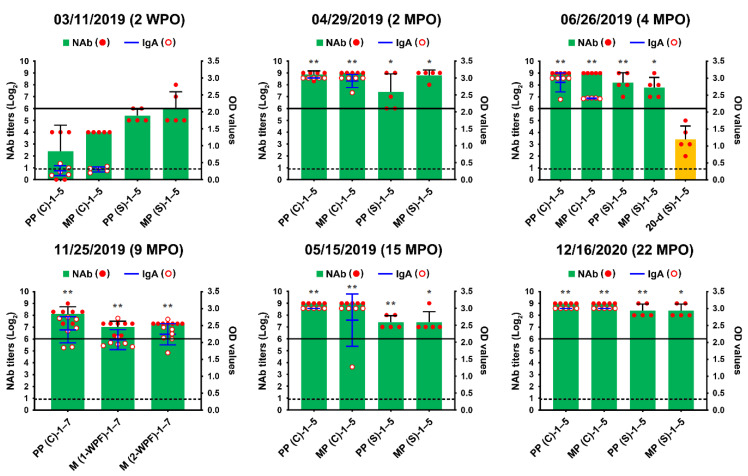
PEDV-specific antibody kinetics in serum (S) and colostrum (C) samples of the sows up to 22 months post-outbreak (MPO). The serum samples were collected from five primiparous (PP-1–5) and five multiparous (MP-1–5) sows or five 20-day-old piglets (20-d-1–5) at the indicated sampling times and subjected to a virus neutralization test (column chart; left *y*-axis). The colostrum samples from the sows (PP-1–5 and MP-1–5) at the indicated sampling times were subjected to a virus neutralization test (column chart; left *y*-axis) and the PEDV-S1 IgA ELISA (line chart; right *y*-axis). On 25 November 2019, the colostrum (C) and milk (M) samples were collected on and after one or two weeks postfarrowing (1- or 2-WPF) from seven primiparous sows and tested using ELISA. Neutralizing antibody (NAb) titers are presented on a log_2_ scale. The hypothetical protective NAb titer (1:64) against PEDV is indicated by a bold line. The samples above the OD cutoff value of 0.3 from the IgA ELISA (dashed line) were considered to be positive. The date of sample collection (month/day/year) is indicated at the top of each graph. Results are expressed as the mean NAb and IgA titers from the same pig group, and error bars represent the mean ± SDM. The results from each group on 11 March 2019, two weeks post-outbreak (2 WPO), before vaccination were compared with those from corresponding groups in the indicated sampling points after L/K/K vaccination. *, *p* < 0.05; **, *p* < 0.001.

**Figure 3 pathogens-10-00830-f003:**
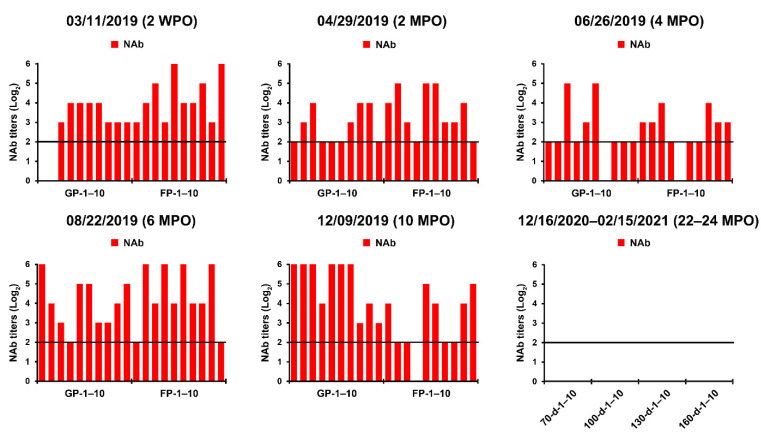
PEDV-specific antibody responses in serum samples of grow/finish pigs up to 24 months post-outbreak (MPO). The samples were collected from 10 growing pigs (GP-1–10; age, 70–100 days) and 10 finishing pigs (FP-1–5; age, 130–160 days) at the indicated sampling times and tested using a virus neutralization test. On 16 December 2020, 14 January 2021, and 15 February 2021, the samples were collected from 40 pigs at 70 days (*n* = 10; 70-d-1–10), 100-day (*n* = 10; 100-d-1–10), 130 days (*n* = 10; 130-d-1–10), and 160 days (*n* = 10; 160-d-1–10) of age and tested. Neutralizing antibody (NAb) titers of individual samples are presented on a log_2_ scale. The samples above the neutralizing endpoint titer 1:4 (dashed line) were considered to be positive. The date of sample collection (month/day/year) is indicated at the top of each graph.

**Figure 4 pathogens-10-00830-f004:**
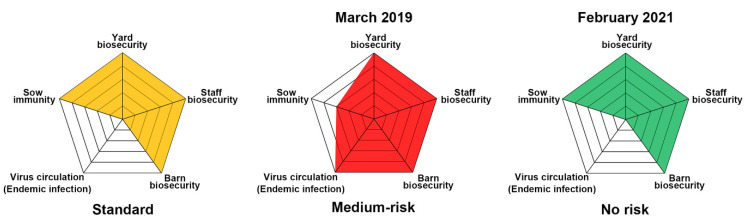
Illustration of the potential risk of PED recurrence determined using the risk pentagon profile diagram. The five parameters assessed in the risk pentagon system include the levels of yard biosecurity, staff biosecurity, barn biosecurity, sow immunity, and virus circulation (endemic infection). Each factor, given a score of 1–5, is represented by a shaded area relative to the pentagon’s total area. The first pentagon diagram (standard) represents a normal model with no risk of the occurrence or recurrence of PED. The results (dates) of each initial (March 2019) and last (February 2021) assessment are indicated as medium or no risk underneath each pentagon.

**Figure 5 pathogens-10-00830-f005:**
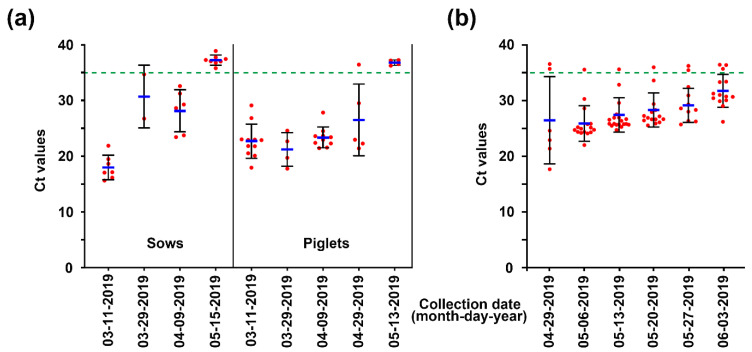
Initial monitoring of PEDV status in individual pigs or pen-level slurries from affected farrowing rooms using rRT-PCR. (**a**) Fecal shedding of PEDV in sows and piglets. The viral load in rectal swab samples collected from sows (left) or piglets (right) at the indicated sampling time point was determined using rRT-PCR analysis. (**b**) PEDV contamination in the farrowing rooms. The viral load in slurry samples collected from each farrowing house at the indicated sampling time point was determined using rRT-PCR analysis. The mean Ct values from all samples at each sampling time point are shown, and error bars denote the mean ± SDM. The samples below the mean Ct value cutoff value of 35 from rRT-PCR (green dashed line) were considered to be positive.

**Figure 6 pathogens-10-00830-f006:**
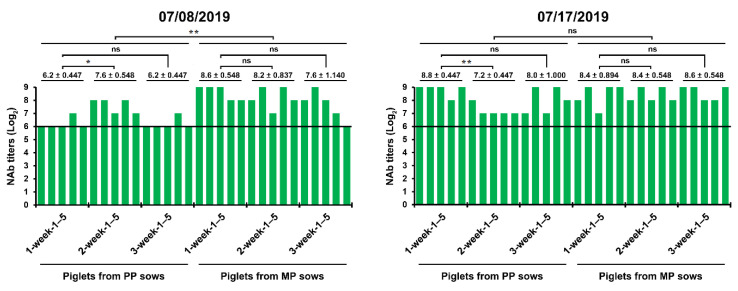
PEDV-specific antibody kinetics in serum samples of the suckling piglets. The samples were collected from 30 piglets 1 (*n* = 10), 2 (*n* = 10), and 3 (*n* = 10) weeks after birth from primiparous (PP) and multiparous (MP) sows at the indicated sampling times and tested using a virus neutralization test. Neutralizing antibody (NAb) titers of individual samples are presented on a log_2_ scale. The hypothetical protective NAb titer (1:64) against PEDV is indicated by a bold line. The date of sample collection (month/day/year) is indicated at the top of each graph. The mean NAb titers (±SDM) from the sample pig group are shown on a log_2_ scale. *, *p* < 0.05; **, *p* < 0.001; ns, not significant.

**Figure 7 pathogens-10-00830-f007:**
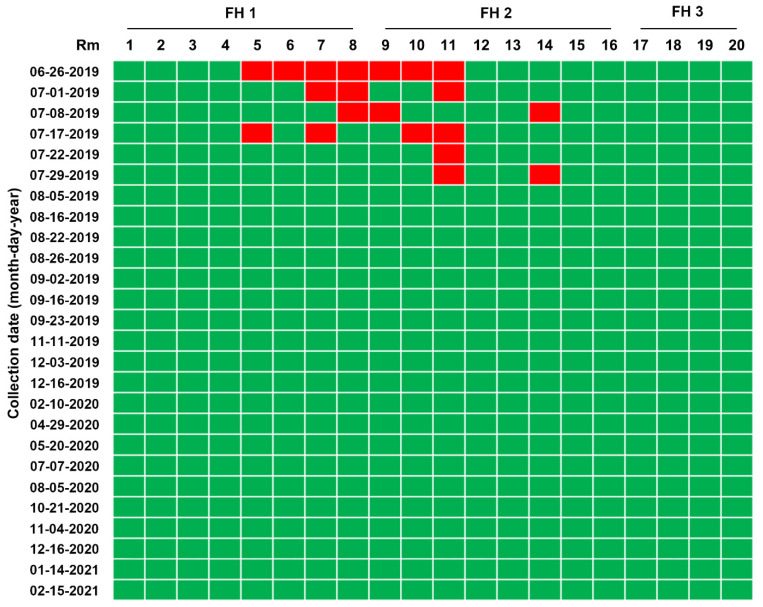
Longitudinal monitoring of PEDV status in pen-level slurry samples from 20 rooms (Rm 1–20) through 3 farrowing houses (FH 1–3) using rRT-PCR. PEDV detection in slurry samples collected from each farrowing barn at the indicated sampling times was determined using rRT-PCR analysis. The samples below the mean Ct value cutoff value of 35 from the rRT-PCR were considered positive. Rooms with PEDV-positive slurries are displayed in red, whereas rooms with PEDV-negative slurries are shown in green.

**Figure 8 pathogens-10-00830-f008:**
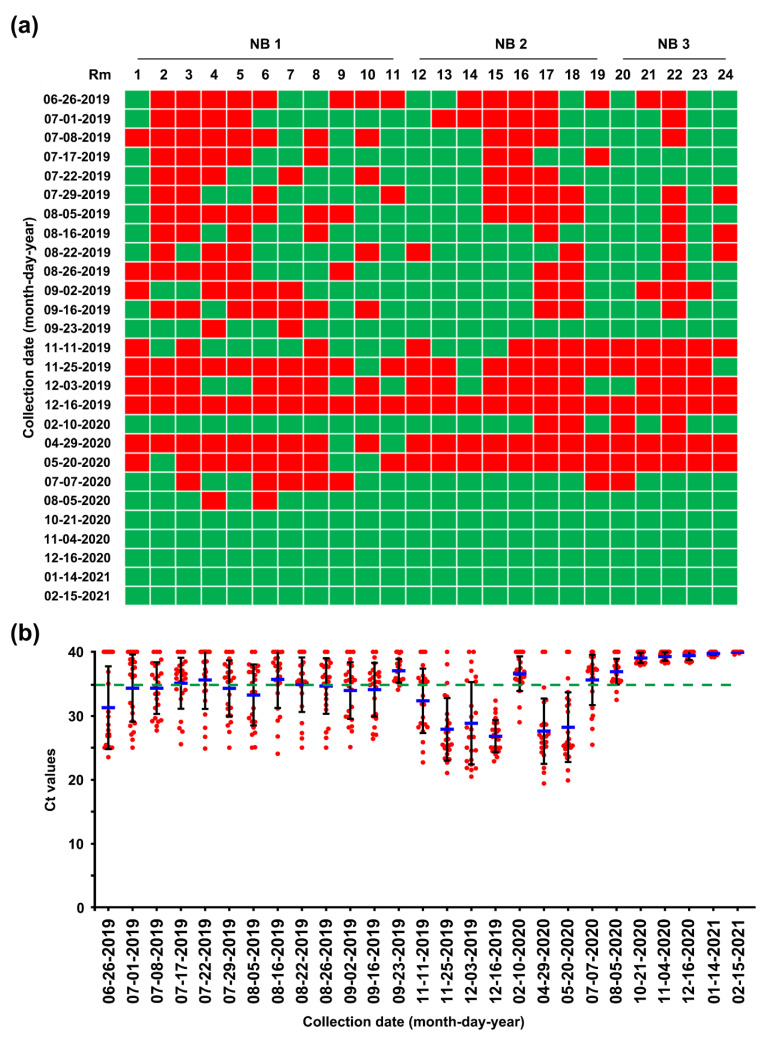
Longitudinal monitoring of PEDV status in pen-level slurry samples from 24 rooms (Rm 1–24) throughout 3 nursery barns (NB 1–3) using rRT-PCR. (**a**) PEDV detection in nursery barns. Pen-level slurry samples were collected from each nursery barn at the indicated sampling times, and PEDV contamination was determined using rRT-PCR analysis. Rooms with PEDV-positive slurries are displayed in red, whereas rooms with PEDV-negative slurries are shown in green. (**b**) Viral load in the slurry samples collected from the nursery barns. The mean Ct values from all samples at each sampling time point are shown, and error bars denote the mean ±SDM. The samples below the mean Ct value cutoff value of 35 from rRT-PCR (green dashed line) were considered to be positive.

**Figure 9 pathogens-10-00830-f009:**
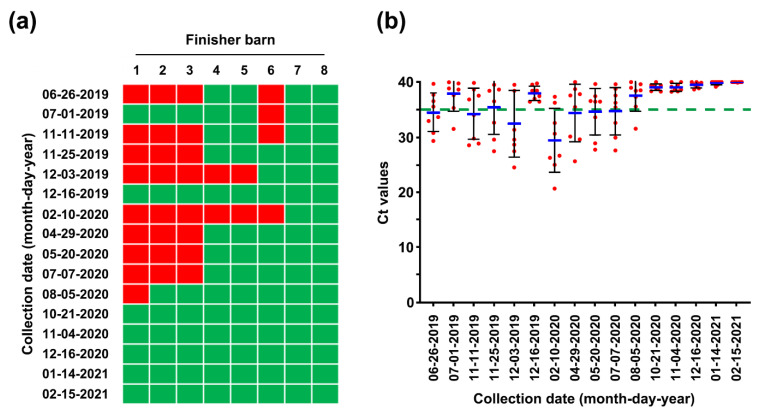
Longitudinal monitoring of PEDV status in pen-level slurry samples from eight finisher barns (1–8) using rRT-PCR. (**a**) PEDV detection in the finisher barns. Pen-level slurry samples were collected from each finisher barn at the indicated sampling times, and PEDV contamination was determined using rRT-PCR analysis. Barns with PEDV-positive slurries are displayed in red, whereas barns with PEDV-negative slurries are shown in green. (**b**) Viral load in the slurry samples collected from the finisher barns. The mean Ct values from all samples at each sampling time point are shown, and error bars denote the mean ± SDM. The samples below the mean Ct value cutoff value of 35 from rRT-PCR (green dashed line) were considered to be positive.

**Figure 10 pathogens-10-00830-f010:**
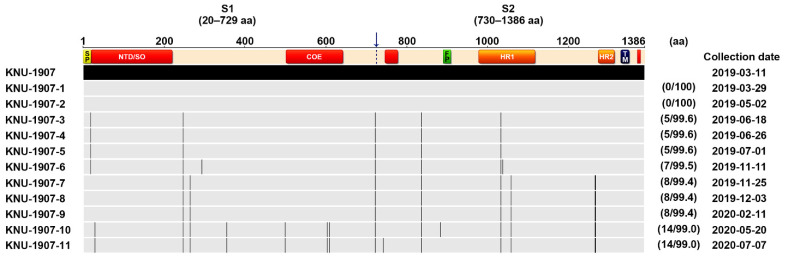
Schematic diagram of genetic variations in the S gene among PEDV strains temporally collected in the study. The top illustration represents the organization of the S protein, featuring the S1 and S2 domains that contain a signal peptide (SP), an N-terminal hypervariable domain (NTD), a fusion peptide (FP), heptad repeat regions (HR1 and HR2), and a transmembrane domain (TM). Red-highlighted areas in the diagram depicting the S protein represent four neutralizing epitopes (NTD/SO, residues 19–220; COE, residues 502–641; residues 744–774; residues 1371–1377) of HP-G2b PEDV. Lightly shaded areas are identical to KNU-1907, and the vertical black bars represent one amino acid sequence that is divergent from that of KNU-1907. The digits in parentheses on the right indicate the number of amino acid changes and the percent identity compared with KNU-1907.

**Figure 11 pathogens-10-00830-f011:**
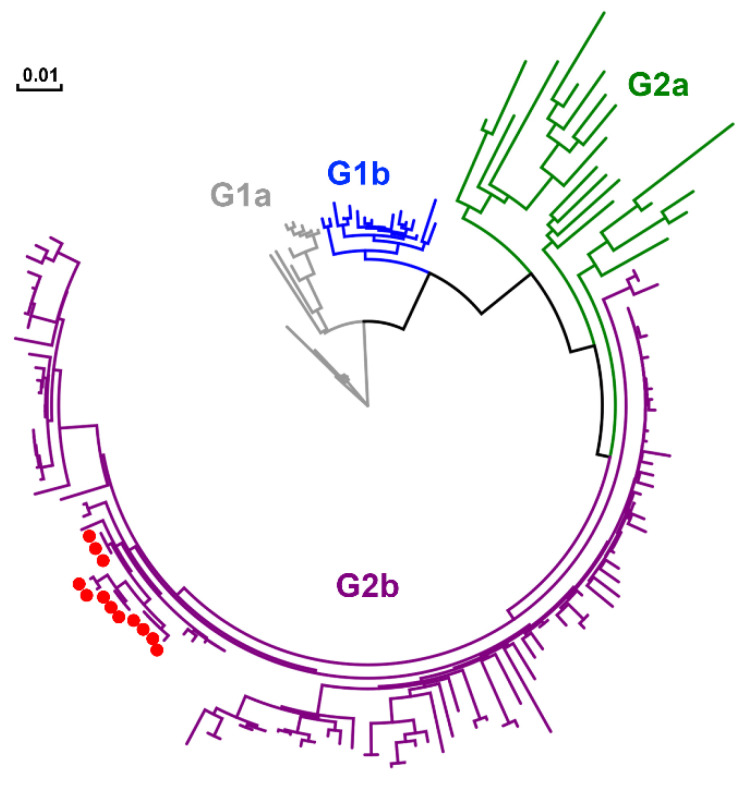
Phylogenetic analysis based on the full S genes of the PEDV strains on the farm. Red dots indicate the KNU-1907 strains temporally identified in this study. Four genotypes, G1a (gray), G1b (blue), G2a (green), and G2b (purple), are indicated. The scale bar indicates the number of nucleotide substitutions per site.

**Figure 12 pathogens-10-00830-f012:**
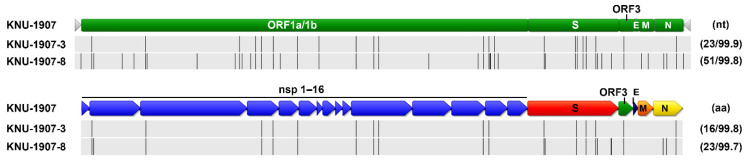
Schematic diagram of the genomic differences among PEDV strains at the nucleotide (nt, top) and amino acid (aa, bottom) levels. The top illustration represents the genomic regions, with green bars symbolizing the identified ORFs. Light gray arrows represent each 5′- and 3′-untranslated region. The second diagram represents the genomic regions, with blue and various arrows indicating the nonstructural (nsp1–16) and structural (S-ORF3-E-M-N) proteins, respectively. Lightly shaded areas are those identical to KNU-1907, and the vertical black bars represent one nucleotide (nt) or amino acid (aa) sequence that is divergent from that of KNU-1907. The digits in parentheses on the right indicate the number of nt or aa changes and the percent identity compared with KNU-1907.

**Figure 13 pathogens-10-00830-f013:**
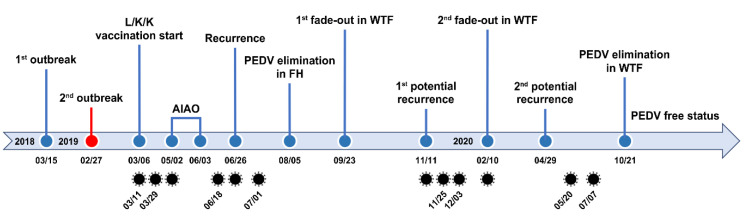
Timeline indicating important events, including PED outbreaks, elimination procedures and measurements, and virus detection, during the elimination study. FH: farrowing houses, WTF: wean-to-finish barns. The diagram illustrating the virus (
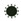
) represents the date of virus detection.

**Figure 14 pathogens-10-00830-f014:**
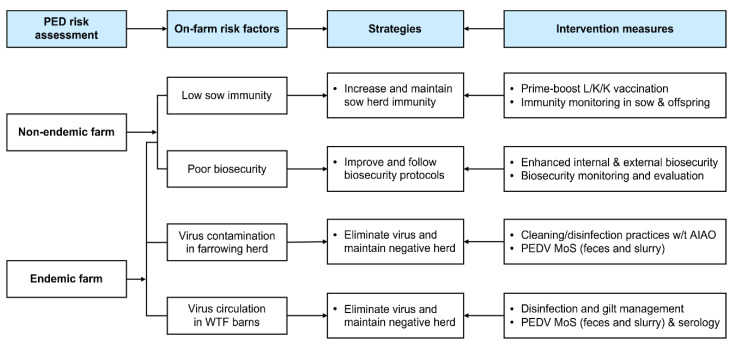
Customized disease control strategies and measures upon PEDV exposure. Once PEDV is introduced into a swine herd, a risk assessment of PED recurrence is conducted to identify on-farm risk factors that should be managed to control PED and prevent re-exposure in the affected farm. The main factors influencing PED recurrence in farrow-to-finish (FTF) farms can be divided into four different categories: low sow immunity, poor biosecurity, virus contamination in farrowing houses, and virus circulation in wean-to-finish (WTF) barns. The four-pillar-based coordinated intervention strategies for disease control are utilized to stimulate the immunity or/and eliminate the virus for eradication in the FTF herd.

**Table 1 pathogens-10-00830-t001:** The information of PEDV spike and full genome sequences obtained during the study.

Detection Date	Source	Strain Name	Sequence
03-11-2019	Feces	KNU-1907	Full genome
03-29-2019	Feces	KNU-1907-1	Spike
05-02-2019	Slurry	KNU-1907-2	Spike
06-18-2019	Feces	KNU-1907-3	Full genome
06-26-2019	Slurry	KNU-1907-4	Spike
07-01-2019	Feces	KNU-1907-5	Spike
11-11-2019	Slurry	KNU-1907-6	Spike
11-25-2019	Slurry	KNU-1907-7	Spike
12-03-2019	Feces	KNU-1907-8	Full genome
02-11-2020	Slurry	KNU-1907-9	Spike
05-20-2020	Slurry	KNU-1907-10	Spike
07-07-2020	Slurry	KNU-1907-11	Spike

**Table 2 pathogens-10-00830-t002:** The evolution of KNU-1907 present in the farm using spike or full genome sequences.

	Substitution	
	Mean (±S.E. ^1^)	95% HPD ^2^
Spike (nt)	1.683 × 10^−4^ (±0.02 10^−4^)	2.082 × 10^−5^ (low) 3.216 × 10^−4^ (high)
Spike (aa)	2.239 × 10^−3^ (±0.02 10^−3^)	4.328 × 10^−4^ (low) 4.215 × 10^−3^ (high)
Full genome	4.921 × 10^−7^ (±0.03 10^−7^)	8.845 × 10^−8^ (low) 8.412 × 10^−7^ (high)

^1^ Standard error. ^2^ 95% highest probability density.

## Data Availability

The data presented in this study are available in the article and its Appendix A.

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
