# Peer review of "Successful Eradication of Porcine Epidemic Diarrhea in an Enzootically Infected Farm: A Two-Year Follow-Up Study"

_pathogens, 2021, doi:10.3390/pathogens10070830_

Round 1
Reviewer 1 Report
General Comment
The study reported Successful Eradication of Porcine Epidemic Diarrhea in An Enzootically Infected farm: A Two-year Follow-up Study
In general, the paper was well written and understandable.
Specific comments
1. The conclusion section should be expanded, for example:
- for other viral diseases, the four pillar-based coordinated intervention measures can also be used
- how this method can be applied to other infectious disease eradication?
Reviewer 2 Report
Brief summary
The manuscript entitled “Successful Eradication of Porcine Epidemic Diarrhea in An Enzootically Infected farm: A Two-year Follow-up Study” described the intervention measures implemented for PEDV elimination in a farm over 2 years period. The authors conducted the risk assessments, serology, and viral genetic analysis. This study method was novel and might be beneficial for the farm management to break PEDV circulation cycle.
Broad comments
Nearly all the data herein is descriptive. The main weakness of this research study is missing control to show the effectiveness of interventions.
There are many grammar errors. For example, in the title, the nouns (farm) should be capitalized, while the article (An) should not be.
This manuscript is too long (23 pages) and difficult to read. Although the summary in the conclusion is helpful, this readability can be improved by summarizing all the timelines and important events in a table and shorten the word contents in the results.
A decision tree can be used to explain the rationale of strategies/measurements applied.
It is extremely difficult to understand the correlation between these dates- should be changed as n days post-outbreak.
The contents in the result are not well-prepared. For example,
Result 2.1. The authors use viral genetic analysis to show the virus KNU-1907 was the original virus on the farm rather than introduced from outside. However, the background of these reference viral strains (KNU-1836, 1904, 1911, 1704, 1708) was not mentioned. Therefore, the readers can not understand how the conclusion is reached. On the other hand, some of the procedures (lines 72-79) should be moved to materials and methods.
** Again, the Discussion section is too long and needs to be condensed.
Specific comments
Line 124-128. The data in this study can’t be directly compared with those from another study (Jang et. al., 2019), given the methods can be different. At the very least, it should be stated in the Discussion, rather than the Results section.
Line 176-178. Please use a statistic test (paired t-test) to show the Ab titer significantly increased after vaccination.
Line 231-235. A statistic test should be applied.
Line 243-255. Many uncertain words (probably, seemed to be) used here are confusing.
Figure 2. A line chart should not be used herein because each pig is an independent individual. Instead, the data from the same pig group can be combined and expressed as mean with variations (dot plot or box) as Figure 5.
Figure 11 is redundant because no virus in this farm changes the genogroup.
Discussion (lin440-443): It is confusing the authors developed an effective new G2b live vaccine but claimed unavailability of an efficient vaccine for HP-G2b PEDV. Is there any cross-protection?
Reviewer 3 Report
The authors describe a PED elimination project in a farrow to finish farm. Although PED elimination is common in places such as North America there are no studies describing the elimination procedures to the level described in this manuscript. Furthermore, most of the elimination projects have been conducted in farrow to wean farms but not farrow to finish barns, so this study is a nice addition to the literature and it will serve as a reference for future studies.
This author just have some minor comments, mostly in regards to terminology:
L126 - what does unstable antibodies mean? A better way to refer to the antibody levels there would be "variable antibody levels"
L178 - same comment as above "stable" titers should be replaced by "unchanged or similar antibody levels" if this is what the authors meant to say
L289- What does virus extirpation mean? Virus elimination? In the opening section of the discussion it would be better to mention that the challenge to eliminate PED in farrow to finish farms is because the continue flow nature of the animals at the farm and the level of environmental contamination. I'm afraid we have the knowledge, the difficulty comes into the implementation of the measures.
Overall this is a very well written manuscript that describes in details on the processes followed during the project.
